# The Impact of Using a Larger Forearm Artery for Percutaneous Coronary Interventions on Hand Strength: A Randomized Controlled Trial

**DOI:** 10.3390/jcm10051099

**Published:** 2021-03-06

**Authors:** Pawel Lewandowski, Anna Zuk, Tomasz Slomski, Pawel Maciejewski, Bogumil Ramotowski, Andrzej Budaj

**Affiliations:** Centre of Postgraduate Medical Education, Cardiology Department, Grochowski Hospital, 04-073 Warsaw, Poland; azuk@cmkp.edu.pl (A.Z.); tomek.slomski@gmail.com (T.S.); pmaciejewski@cmkp.edu.pl (P.M.); bramotowski@cmkp.edu.pl (B.R.); abudaj@cmkp.edu.pl (A.B.)

**Keywords:** complications, hand grip, transradial access, transulnar access

## Abstract

(1) Background: The exact mechanism underlying hand strength reduction (HSR) after coronary angiography with transradial access (TRA) or transulnar access (TUA) remains unknown. (2) Methods: This study aimed to assess the impact of using a larger or smaller forearm artery access on the incidence of HSR at 30-day follow-up. This was a prospective randomized trial including patients referred for elective coronary angiography or percutaneous coronary intervention. Based on the pre-procedural ultrasound examination, the larger artery was identified. Patients were randomized to larger radial artery (RA) or ulnar artery (UA) or a group with smaller RA/UA. The primary endpoint was the incidence of HSR, while the secondary endpoint was the incidence of subjective HSR, paresthesia, and any hand pain. (3) Results: We enrolled 200 patients (107 men and 93 women; mean age 68 ± 8 years) between 2017 and 2018. Due to crossover between TRA and TUA, there were 57% (*n* = 115) patients in larger RA/UA and 43% (*n* = 85) patients in smaller RA/UA. HSR occurred in 29% (*n* = 33) patients in larger RA/UA and 47% (*n* = 40) patients in smaller RA/UA (*p* = 0.008). Subjective HSR was observed in 10% (*n* = 12) patients in larger RA/UA and 21% (*n* = 18) patients in smaller RA/UA (*p* = 0.03). Finally, paresthesia was noted in 7% (*n* = 8) patients in larger RA/UA and 22% (*n* = 15) in smaller RA/UA (*p* = 002). Independent factors of HSR were larger RA/UA (OR 0.45; 95% CI, 0.24–0.82; *p* < 0.01) and the use of TRA (OR 1.87; 95% CI, 1.01–34; *p* < 0.05). (4) Conclusions: The use of a larger artery as vascular access was associated with a lower incidence of HSR at 30-day follow-up.

## 1. Introduction

The radial artery (RA) has become the preferred vascular access in cardiovascular interventions [1,2,3]. In comparison with transfemoral access (TFA), transradial access (TRA) is related to the reduction in the incidence of vascular complications and mortality after a percutaneous coronary intervention (PCI) among patients with acute coronary syndrome (ACS) [2]. The ulnar artery (UA) and RA are the terminal branches of the brachial artery. The transulnar access (TUA) could offer an alternative to TRA. TUA may be as effective and safe as TRA, especially for experienced operators, when the RA is narrow, with anatomical abnormalities, or the impalpable pulse [4,5,6,7,8,9]. However, this vascular access still needs to be popularized among invasive cardiologists. Although TRA has been increasingly adopted as the main site of vascular access, this vascular access has some limitations, which include an increase in the radiation dose exposure for operators and patients, particularly the inexperienced operators, and a long learning curve [10,11]. Further, challenges with arterial anatomical variations in the forearm, [12] radial artery spasms (RAS) [13] and RA occlusion [14] are among the most frequent causes of TRA failure and local complications. With the increasing use of TRA in invasive cardiology, the interest in possible mechanisms of upper limb dysfunction following invasive procedures has been still growing. Several reasons are mentioned, including neurovascular injuries, endothelial dysfunction, and nerve damage [15,16,17,18,19]. 

Medical hand grip dynamometers are routinely used for the evaluation of muscular, neurological, or skeletal illnesses. Hand grip strength can be easily, reliably, and quantitatively measured using a hand dynamometer [20]. In the Hand Grip test After Transradial (HANGAR) study, the hand dynamometer has also been applied in patients who underwent transradial PCI to assess the impact of RA occlusion on the hand strength [21]. 

There were no studies that analyzed the size of forearm artery access and its impact on the frequency of incidents of hand strength reduction (HSR). 

In this study, we aimed to assess the impact of using a larger forearm artery vs. a smaller artery on the incidence of HSR among patients scheduled for elective coronary angiography (CAG) and PCI with TRA and TUA.

## 2. Material and Methods

### 2.1. Study Design

This single-center prospective randomized trial was conducted between 2017 and 2018. Patients referred to a cath lab or elective CAG or PCI were included in the study. Regardless of sex, patients who were older than 18 years of age and hospitalized for elective CAG were eligible for inclusion in the trial. Patients with a history of previous vascular interventions were also included. Exclusion criteria were age younger than 18 years, hemodialysis, nonpalpable pulse over the RA or UA, ST-segment elevation myocardial infarction (STEMI)**,** non-ST-segment elevation myocardial infarction (NSTEMI), coronary artery bypass graft surgery (CABG) with a bilateral left internal mammary artery or right internal mammary artery or forearm arteries, RA or UA occlusion confirmed by ultrasonography, and a similar dimension of the RA and UA. The clinical trial registration number was DRKS00012923.

A pre-procedural ultrasound examination of the right and left forearm arteries was performed by two experienced sonographers, using the US scanner EUB 5000 (Hitachi, Ltd., Tokyo, Japan) with a 5–10 MHz linear probe in B-mode in a transverse projection. Based on the ultrasound examinations, a larger-diameter artery was identified. An index of artery domination (I×D) was established as follows: an arithmetic average of 3 results of measurements of diameters in 3 levels of the forearm; a wrist level at the point of the perceptible pulse, at half the length of the forearm, and below the elbow. Based on I×D, patients were assigned to a larger UA group (group A) or a larger RA group (group B). After the ultrasound examination, patients were randomly assigned to subgroups where the invasive procedure was performed through a larger UA or a smaller RA (group A). In group B, patients were randomized to a larger RA or smaller UA. For the purpose of statistical analysis, two main groups were created: a larger RA/UA and a smaller RA/UA. The randomization of patients is presented in Figure 1.

Experienced certified interventional cardiologists performed the CAG or PCI, and they were experts in both RA and UA cannulation. Standard 6-Fr radial introducers (Radial Introducer Sheath, Demax Medical Co. Ltd., Sydney, Australia), 6- or 5-Fr diagnostic catheters (Angiodyn, B Braun, Melsungen, Germany), and 5- or 6-Fr guiding catheters (Luncher, Medtronic, Danvers, MA, USA) were used. If the primary access failed, the next vascular access was an ipsilateral vessel RA or UA. After the vascular access was established, a bolus of unfractionated heparin 5000 IU was injected. In the case of PCI, the total heparin dose was 70–100 IU/kg. Intra-arterial nitroglycerine bolus was administered to prevent vascular spasm. According to standard and local protocols, CAG was performed along with PCI ad hoc, if needed. To achieve hemostasis, a standard RA gauze compression for at least 2 h afterward was used.

#### 2.1.1. Hand Grip Strength Measurements

The hand grip strength was assessed with the electronic MG4800 medical hand grip dynamometer (Charder Electronic Co., Ltd., Taichung City, Taiwan) using the well-established Southampton protocol, based on the American Society of Hand Therapist recommendations [22]. The patient was sitting comfortably in a standard chair with fixed arms on the table in a quiet examination room. The temperature range was 21–22 °C. For every measurement, the same devices in the same environment were used for all participants. The patients were asked to rest their forearms on the table with the wrist just over the end of the edge of the table and positioned the wrist in a neutral position with the thumb facing upward. The observer encouraged the patient to squeeze as long and strong as possible, starting with the right hand. In the same position of the body, the maximum strength of the hand and mean strength of five grips were measured and expressed in kilograms (kg). The measurements were done in the left and right hand. After 30 days, the whole procedure of hand strength measurements was repeated. The differences of hand strength between pre-procedural values and 30-day values were used for the statistical analysis. The observer measuring and recording maximum and medium hand strength values after 30 days of follow-up was blinded to the allocation process.

#### 2.1.2. Hand Strength Reduction Incidence

The primary endpoint of the study was defined as the incidence of strength reduction in the used hand for CAG/PCI at 30-day follow-up in larger RA/UA and smaller RA/UA. 

#### 2.1.3. Additional Complications 

The secondary endpoint was defined as the incidence of paresthesia, any pain of used hand, and a patient-subjective reduction in the strength of the used hand at 30-day follow-up. 

A visual analog scale (VAS) was used to evaluate patients’ pain associated with both larger RA/UA vs. smaller RA/UA [23]. VAS was performed immediately after CAG/PCI and at 24-h observation after CAG/PCI. 

The impact of the use of larger RA/UA vs. smaller RA/UA on the efficacy and safety of vascular access was presented in a separate article [24].

The trial was approved by the Centre of Postgraduate Medical Education Bioethical Committee and conducted in accordance with the Declaration of Helsinki (32/PB/2017; 12 April 2017). All patients provided written informed consent.

### 2.2. Statistical Analysis

Baseline demographic characteristics and clinical data were presented as means and standard deviations (SD). Categorical data were shown as frequencies, compared with chi-square test statistics or Fisher’s exact test. Multivariable logistic regression analysis was performed to identify independent predictors for endpoints at the significance level of 5%. Odds ratios (OR) with 95% confidence intervals (CI) were provided for significant predictors. Statistical analysis for the cut-off of average diameters (I×D of RA, I×D of UA) was performed using ROC curve: for the whole group *n* = 200 (for combined RA and UA), and the RA and UA separately (*n* = 100), as on-treatment analysis. STATA v.14 software for Windows was used for statistical calculations (STATA Corporation, College Station, TX, USA).

## 3. Results

Overall, 200 patients (107 men and 93 women; mean age 68 ± 8 years) were randomized into four groups: group A with larger UA (*n* = 50) and smaller RA (*n* = 50), and group B with larger RA (*n* = 50) and smaller UA (*n* = 50). After crossover events between the RA access and UA access, the larger artery group (RA or UA) and smaller artery group (RA or UA) ultimately included 58% (*n* = 115) and 42% (*n* = 85) of patients, respectively. The main indication of CAG was a suspected coronary artery disease (CAD) (Table 1). There were no statistically significant differences between the two groups. The baseline characteristics of patients are presented in Table 1. 

For arterial access, the right limb was used in 88% of patients (*n* = 101) in larger RA/UA and in 80% of patients (*n* = 70) in smaller RA/UA (*p* = 0.2). PCI was performed in 10% of patients (*n* = 11) in larger RA/UA and in 4% of patients (*n* = 3) in smaller RA/UA (*p* < 0.05) (Table 1). Other periprocedural data and concomitant medications are shown in Table 1. 

### 3.1. Hand Strength Reduction 

The incidents of used HSR were observed statistically significantly more frequently in smaller RA/UA 47 % (*n* = 40) than in larger RA/UA 29% (*n* = 33); *p* = 0.008 (Table 2).

### 3.2. Additional Complications

Number of patients with complaints of paresthesia was greater in smaller RA/UA 22% (*n* = 15) than in larger RA/UA 8% (*n* = 7); *p* = 0.002. In addition, subjective HSR was more frequent in smaller RA/UA than in larger RA/UA. After 30-day observation, there were no differences between both groups in the frequency of events of hand pain (Table 2).

### 3.3. Visual Analog Scale for Pain

In the analysis of VAS at the procedure, the pain was greater in smaller RA/UA than in larger RA/UA (*p* < 0.048) (Table 3). There was no difference in VAS outcomes after 24 h of observation.

### 3.4. Changes in the Hand Grip Value

Comparing hand grip strength (maximum and mean value) among patients from larger RA/UA and smaller RA/UA, there were no statistically significant differences in the used and also in the unused hand at the baseline (pre-procedural measurements) (Table 4).

At the 30-day follow-up, there was a reduction in the maximum and mean value of the hand strength of the used hand in the smaller RA/UA group. The HSR in the smaller RA/UA group was statistically significantly greater compared to that in larger RA/UA (change in the maximum value (kg): larger RA/UA 0.46 ± 4.5; smaller RA/UA −1.56 ± 6.21; *p* < 0.001, change in the mean value (kg): larger RA/UA 0.61 ± 4.57; smaller RA/UA −0.79 ± 4.78; *p* = 0.008) (Figure 2 and Figure 3). 

The independent factors for HSR after 30 days of the observation were as follows: access by smaller artery for intervention and RA use (Table 5). 

Based on the univariate logistic regression, the change in mean diameter (I×D) by 1 unit (1 mm) of the RA or UA was associated with the odds ratio of HSR (average value) at 30-day follow-up. The predictive value of arterial diameter (I×D) for HSR was of borderline statistical significance for combined RA and UA groups. (OR 0.53; 95% CI 0.27–1.02; *p* = 0.058). However, separate analyses for RA and UA did not reveal statistical significance.

## 4. Discussion

This single-center randomized study showed that using a larger forearm artery for percutaneous coronary interventions reduces the frequency of episodes of strength reduction in the upper limb. Additionally, it has an impact on reducing the level of pain during the invasive procedure and the frequency of paresthesia occurrence after the procedure. 

### 4.1. Hand Strength Reduction

#### 4.1.1. Potential Mechanisms of the Hand Disorder after CAG/PCI

The exact mechanism underlying motor or sensory dysfunction of the upper limb following TRA or TUA for CAG/PCI has not been fully elucidated. There are numerous possible explanations. Several muscles, the flexor carpi radialis and flexor carpi ulnaris muscles, flexor pollicis longus tendon, and median or ulnar nerves, lie next to the RA or UA at the wrist. The wrist might have numerous anatomical variants, including muscles, tendons, and vascular structures [25]. These structures can be directly injured during cannulation of the RA or UA. Direct demerging and hematoma result in edema, inflammatory reactions, and secondary compression of the underlying structures. In turn, these factors can lead to motor and sensory deficits. Additionally, the pressure of hemostatic devices to maintain hemostasis may result in transient or permanent ischemia of the main nerves or branches, resulting in motor or sensory deficits [15,21,26,27,28,29]. In a recent meta-analysis, there are about 15 studies on TRA involving 3616 participants. The authors of three of these studies reported nerve damage with a combined incidence of 0.16%. The other four studies described sensory loss, tingling, and numbness with a pooled incidence of 1.61%. Pain after TRA was the most common form of limb dysfunction (7.77%) reported in the other three studies. On the other hand, the incidence of hand dysfunction, defined as disability, grip strength change, power loss, or neuropathy, was low (0.49%) [19].

#### 4.1.2. Outcomes of Previous Studies 

The RADAR trial investigators evaluated the hand grip test according to the Allen test’s (AT) outcomes for blood flow. In total, 206 patients after CAG or PCI were assigned to three groups (with different outcomes of AT) in which hand grip tests did not differ between all groups at 24-h, 1-month, and 1-year follow-up [30]. In the HANGAR trial, among patients with RA occlusion after CAG or PCI with TRA, there was a significant reduction in hand grip strength the day after; however, this reduction was no longer observed at the follow-up examination. In our study, there were statistically significant differences in hand grip strength reduction between larger RA/UA and smaller RA/UA. According to our data at 30-day follow-up, there were no significant differences in the unused hand grip strength compared with that at baseline.

#### 4.1.3. Pre-Procedural Selection of the Forearm Artery

The most common complaint concerning the used hand was pain and paresthesia. The frequency of subjective reduction in the hand strength was lower in the larger RA/UA group than in the smaller RA/UA group. The reason for this reduction in hand strength of the used hand was probably due to procedural factors and local complications (RA or UA occlusion, hematoma, pseudoaneurysm, arteriovenous fistula). Our findings support using the pre-procedural ultrasonography to select the best forearm artery for vascular access to reduce the frequency of hand dysfunction. Knowledge of the arterial anatomy, size of arteries, any other abnormalities (calcifications, occlusions) before invasive procedures can be helpful in avoiding complications. Choosing an artery larger in diameter may also have an effect on reducing the incidence of cramps and pain experienced by patients. Recently, in many cath labs, invasive cardiologists have started using the distal part of the RA as vascular access. Furthermore, in this approach, a pre-procedural ultrasound examination could be an excellent tool to evaluate the quality of the artery and patency of the deep and superficial arch of the hand, as well as to facilitate cannulation.

### 4.2. Impact on Daily Practice

Based on multivariate regression analysis outcomes, our study presented several practical results. In patients undergoing CAG/PCI, using a larger forearm artery (RA or UA) as artery access is associated with a lower probability of HSR. Additionally, the use of the TRA is associated with a higher rate of hand grip reduction than TUA at the 30-day follow-up. These data may be important for interventional cardiologists and their patients using their hands for very precise tasks.

### 4.3. Limitations

We enrolled patients after earlier CAG or PCI, which might have impacted the results of the pre-procedural ultrasound examination. However, these patients constitute a large part of the population in cath labs. Our research was designed as a single-center study with a limited number of patients, and it cannot be excluded that an increased number of enrolled patients would have yielded different results. The dynamometer applied in our study has not been previously used in this kind of patient. To diagnose muscle or nerve disability, we could use more specific examinations, such as electromyography and electroneurography. To avoid bias in the reporting of paresthesia, standardized questionnaires, such as the Michigan Hand Questionnaire (MHQ), Carpal Tunnel Questionnaire (CTQ), should be used. 

## 5. Conclusions

Access by the larger artery was associated with a lower incidence of HSR. A diameter of forearm arteries and use of TRA were revealed as the main independent factors impacting the HSR. 

## Figures and Tables

**Figure 1 jcm-10-01099-f001:**
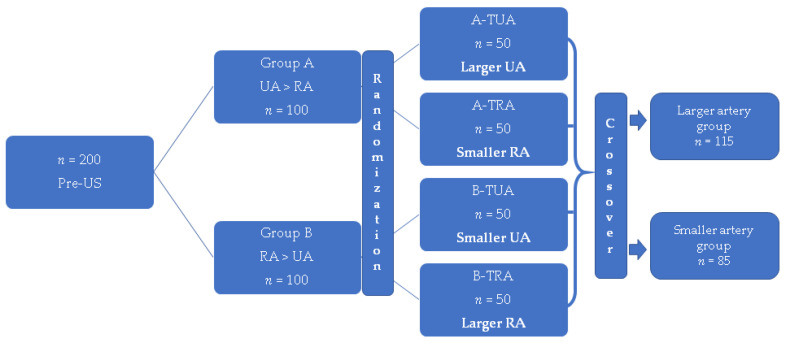
Stratified randomization of patients into larger or smaller artery groups. Pre-US–pre-procedural ultrasonography, RA–radial artery, UA–ulnar artery, TUA–transulnar access, TRA–transradial access

**Figure 2 jcm-10-01099-f002:**
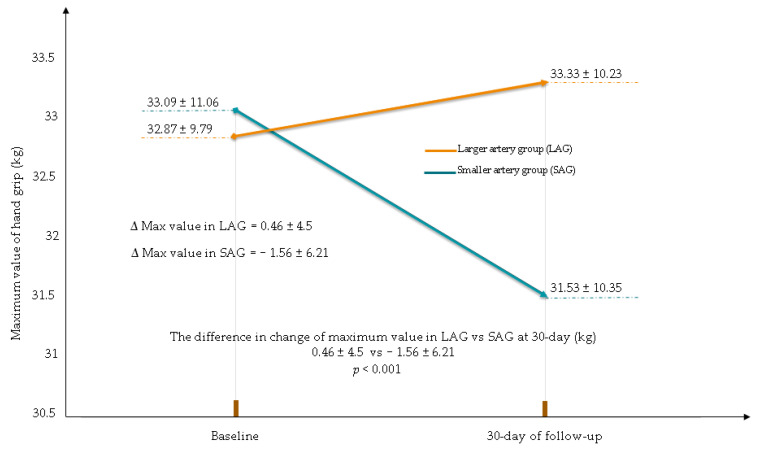
Maximum value reduction in hand grip in used hand for CAG/PCI. CAG—coronary angiography, PCI—percutaneous coronary interventions.

**Figure 3 jcm-10-01099-f003:**
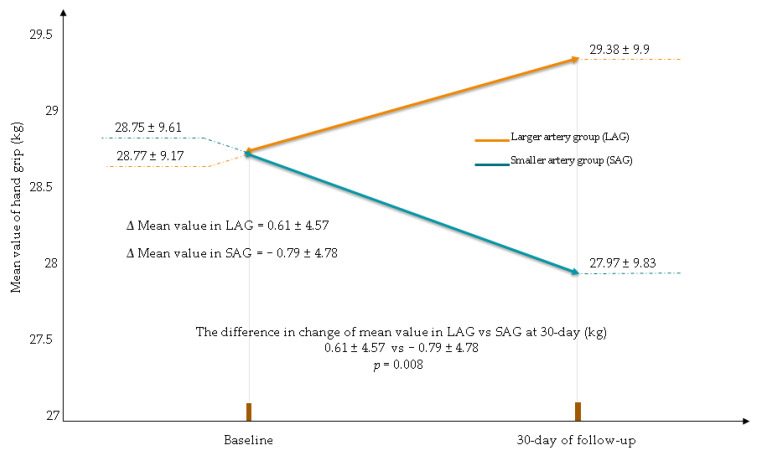
Mean value reduction in hand grip in used hand for CAG/PCI. CAG—coronary angiography, PCI—percutaneous coronary interventions.

**Table 1 jcm-10-01099-t001:** Demographic and clinical characteristics of patients, periprocedural characteristics of coronary angiography, and concomitant use of medications.

	Larger UA/RA (*n* = 115)	Smaller UA/RA (*n* = 85)	*p*-Value
Age, years (mean ± SD)	68 ± 8	68.5 ± 7	0.45
Male, *n* (%)	64 (56)	43 (51)	0.5
BMI, kg/m^2^ (mean ± SD)	28.5 ± 4.6	28.5 ± 5.5	0.73
BSA, m^2^ (mean ± SD)	1.95 ± 0.25	1.94 ± 0.2	0.95
Medical history			
Smoking, *n* (%)	28 (24)	24 (28)	0.53
Hypertension, *n* (%)	113 (98)	82 (96)	0.65
Hypercholesterolemia, *n* (%)	112 (97)	79 (92)	0.17
Peripheral artery disease, *n* (%)	23 (20)	11 (13)	0.19
Diabetes, *n* (%)	38 (33)	24 (28)	0.46
Stroke, *n* (%)	4 (3)	4 (5)	0.72
Renal insufficiency, *n* (%)	20 (17)	10 (12)	0.27
Myocardial Infarction, *n* (%)	22 (19)	14 (16)	0.62
CABG, *n* (%)	4 (3)	3 (4)	1.0
Prior CAG or PCI, *n* (%)	28 (29)	21 (25)	0.53
Diagnosis upon admission			
Suspected CAD, *n* (%)	106 (92)	78 (92)	0.9
CHF, *n* (%)	3 (3)	2 (2)	1.0
Ventricular arrythmia, *n* (%)	6 (5)	5 (6)	0.83
Pre-procedural medications			
Aspirin, *n* (%)	112 (97)	84 (99)	0.63
Clopidogrel, *n* (%)	110 (96)	82 (96)	1.0
Warfarin, *n* (%)	9 (8)	4 (5)	0.56
NOAC, *n* (%)	6 (5)	6 (7)	0.58
Statin, *n* (%)	107 (93)	80 (94)	0.76
β-Blocker, *n* (%)	107 (93)	77 (91)	0.52
Angiography alone, *n* (%)	68 (59)	57 (67)	0.25
Angiography and FFR, *n* (%)	5 (4)	1 (1)	0.19
PCI ad hoc, *n* (%)	31 (27)	24 (28)	0.84
Elective PCI, *n* (%)	11 (10)	3 (4)	0.98
TRA, *n* (%)	58 (50.4)	43 (50.6)	0.91
TUA, *n* (%)	57 (49.6)	42 (50.4)	0.98
Right radial or ulnar access, *n* (%)	101 (88)	70 (82)	0.27
Left radial or ulnar access, *n* (%)	14 (12)	15 (18)	0.27
Fluoroscopy time (min) (mean ± SD)	5.4 ± 5.2	4.9 ± 4.2	0.49
Contrast medium (mL) (mean ± SD)	149 ± 94	143 ± 41	0.63
Radiation dose of X-ray (mSv) (mean ± SD)	281 ± 281	246 ± 197	0.64
Time of compression, (min) (mean ± SD)	147 ± 31	149 ± 33	0.49
Nitroglycerin (dose 200 ug) ia, *n* (%)	115 (100)	82 (96)	1.0
Dose of heparin (IU) (mean ± SD)	6008 ± 1600	5900 ± 1544	0.6
Arterial sheath size			
6-Fr, *n* (%)	115 (100)	85 (100)	1.0
Diagnostic catheter size			
6-Fr, *n* (%)	105 (91)	80 (94)	0.45
5-Fr, *n* (%)	2 (1.8)	2 (2)	
Catheter used for PCI, *n* (%)	47 (40)	32 (37)	
6-Fr, *n* (%)	47 (100)	32 (100)	1.0

BMI—body mass index, BSA—body surface index, CABG—coronary artery bypass grafting, CAD—coronary artery disease, CAG—coronary angiography, CHF—congestive heart failure, FFR—fractional flow reserved, NOAC—non-vitamin K antagonist oral anticoagulants PCI—percutaneous coronary intervention, TRA—transradial access, TUA—transulnar access.

**Table 2 jcm-10-01099-t002:** Primary and secondary endpoints after coronary angiography/percutaneous coronary intervention (CAG/PCI) at 30-day follow-up.

	Larger RA/UA (*n* = 115)	Smaller RA/UA (*n* = 85)	*p*-Value
Strength reduction in used hand, *n* (%) †	33 (29)	40 (47)	0.008
Paresthesia of the upper limb, *n* (%)	8 (7)	15 (22)	0.002
Strength reduction—subjective change, *n* (%)	12 (10)	18 (21)	0.03
Any pain of the upper limb, *n* (%)	14 (12)	17 (20)	0.1

Data are presented as number (%) of patients. † primary endpoint. CAG—coronary angiography, PCI—percutaneous coronary interventions, RA—radial artery, UA—ulnar artery.

**Table 3 jcm-10-01099-t003:** Outcomes of visual analog scale (VAS) for pain at the procedure and after 24 h.

24 h of Follow-Up	Larger RA/UA (*n* = 115)	Smaller RA/UA (*n* = 85)	*p*-Value
VAS at the procedure			
(points 0–10) (mean ± SD)	2.63 ± 1.6	3.08 ± 1.8	<0.048
VAS at 24 h after CAG/PCI			
(points 0–10) (mean ± SD)	1.9 ± 2	2.5 ± 2.5	0.08

RA—radial artery, UA—ulnar artery, VAS—visual analog scale.

**Table 4 jcm-10-01099-t004:** Hand grip values and changes in the hand grip values at the baseline and at 30-day follow-up. Data of used and unused hand for coronary angiography CAG/PCI.

	Larger RA/UA (*n* = 115)	Smaller RA/UA (*n* = 85)	*p*-Value
Hand grip of used hand			
Maximum value at the baseline (kg) (mean ± SD)	32.87 ± 9.79	33.09 ± 11.06	0.87
Maximum value at 30-day (kg) (mean ± SD)	33.33 ± 10.23	31.53 ± 10.35	0.22
The difference in change in maximum value at 30-day (kg) (mean ± SD)	0.46 ± 4.5	−1.56 ± 6.21	0.001
Mean value at the baseline (kg) (mean ± SD)	28.77 ± 9.17	28.75 ± 9.61	0.99
Mean value at the 30-day (kg) (mean ± SD)	29.38 ± 9.9	27.97 ± 9.83	0.31
The difference in change in mean value at 30-day (kg) (mean ± SD)	0.61 ± 4.57	−0.79 ± 4.78	0.008
Hand grip of unused hand			
Maximum value at the baseline (kg) (mean ± SD)	30.01 ± 10.01	29.82 ± 9.92	0.74
Maximum value at 30-day (kg) (mean ± SD)	30.31 ± 9.70	30.15 ± 10.22	0.9
The difference in change in maximum value at 30-day (kg) (mean ± SD)	0.3 ± 3.08	0.33 ± 1.95	0.87
Mean value at the baseline (kg) (mean ± SD)	26.38 ± 9.29	26.41 ± 9.19	0.98
Mean value at the 30-day (kg) (mean ± SD)	26.4 ± 9.18	26.73 ± 9.22	0.8
The difference in change in mean value at 30-day (kg) (mean ± SD)	0.01 ± 3.37	0.32 ± 1.94	0.59

CAG—coronary angiography, PCI—percutaneous coronary interventions, RA—radial artery, UA—ulnar artery.

**Table 5 jcm-10-01099-t005:** Logistic regression analysis: independent factors for strength reduction in used hand for CAG/PCI.

	OR (95% CI)	*p*-Value
Larger RA/UA	0.45 (0.24–0.82)	0.01
Use of TRA	1.87 (1.01–3.4)	0.045

CAG—coronary angiography, PCI—percutaneous coronary interventions, RA—radial artery, UA—ulnar artery, TRA—transradial access.

## Data Availability

The data presented in this study are openly available at DOI [https://doi.org/10.3390/jcm10051099].

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
