# Peer review of "The Impact of Using a Larger Forearm Artery for Percutaneous Coronary Interventions on Hand Strength: A Randomized Controlled Trial"

_jcm, 2021, doi:10.3390/jcm10051099_

Round 1
Reviewer 1 Report
This paper from Lewandoski et al is similar to a prior paper from the same population but reports on hand strength reduction rather than spasm or vascular complications. However, overall this report is of interest as it demonstrates that selection of the smaller of the forearm arteries (which I am not sure anyone would do clinically) is associated with greater hand strength reductions than using the larger arteries.
The authors use reasonably validated tools. It might have been better if the observers measuring/recording the hand strength were blinded to the allocation, if this is the case the authors should report as such. The findings are of clinical utility as it helps justify the routine use of ultrasound sizing to determine whether radial or ulnar access should be used.
Reviewer 2 Report
In the manuscript "The impact of use a larger forearm artery for percutaneous coronary interventions on the hand strength: a randomized controlled trial, Lewandowski et al present the results of a randomized controlled trial randomizing coronary angiography and/or intervention to either the larger or smaller forearm artery and 30-day handgrip test was measured. In the 200 patients enrolled, handgrip strength was significantly lower in patients whose smaller forearm artery was used compared to those whose larger artery was used. The design is novel, however the conclusions conflict with existing data such as RADAR and HANGAR. A few comments:
1) Was the diameter of the RA or UA measured? If so, analysis for cutoffs predicting reduced handgrip strength should be highlighted rather than simply measuring relative diameter to the other artery.
2) Please reconcile why in RADAR and HANGAR all recovered their baseline strength on follow up but strength is low in the current study at 30 days. Was it because most of the catheters were 6F? Was there a technique issue? Was immediate post-procedure handgrip strength measured?
3) Tables 1 and 2 could be combined, same with 3 and 5. Table 5 has no follow-up handgrip strength.
Reviewer 3 Report
The authors mention about the clinical impact of use of larger forearm artery for percutaneous coronary intervention in their randomized control trial. It is interesting report. I suggest some point to improve the value of their manuscript.
1) They often use not only radial artery but also ulnar artery. However, trans ulnar access is not so common compared to trans radial access. Is it my misunderstanding? trans-ulnar access itself is not established procedure. They have to mention about it.
2) They reported 29% hand strength reduction was observed in total court. Personally, it is relatively high compared to my experience and previously reported data. Why did they experience such a kind of high complication rate? Please qualify.
3) Same concern to question 2. They reported paresthesia in 7% (larger RA/UA cohort) and 22% (smaller RA/UA cohort). It is also high. approximately 10% complication rate is not acceptable please clarify.
4) Recently, the technique of distal radial access become common. They should mention about it during discussion section.
Round 2
Reviewer 2 Report
All issues addressed. Please site the univariate regression for RA or UA diameter in predicting reduction in handgrip strength in the results (separate figure not needed, just cite the results).
Author Response
We would like to thank the Reviewer for the interest in our manuscript. We believe the process of revision of this manuscript made it much better.
Point-by-point responses to the Reviewers’ comments are provided below. For clarity, our responses are indicated in italics, and textual additions (manuscript changes) are marked by the yellow background color. Reviewer comments are presented in bold text.
Thank you for your consideration. I look forward to hearing from you.
Sincerely,
Pawel Lewandowski
All issues addressed. Please site the univariate regression for RA or UA diameter in predicting reduction in handgrip strength in the results (separate figure not needed, just cite the results).
Response to the Reviewer. According to the Reviewer's recommendation, we added information of univariate regression analyses for RA or UA diameter in predicting a reduction in handgrip strength in the statistical methods section and in the result section.
2.2. Statistical methods
The characteristics of patients were presented as means and standard deviations (SD) for continuous data and frequency tables for categorical data. The primary endpoint outcomes were assessed using an on-treatment analysis. Multiple logistic regression models were fitted to identify independent predictors for binary endpoints at the statistical significance level of 5%. Odds ratios (ORs) with 95% confidence intervals (CIs) were provided for significant predictors. Statistical analysis for cut-off of average diameters (IxD of RA, IxD of UA) was performed using ROC curve: for the whole group n=200 (for combined RA and UA), and for the RA and UA separately (n=100), as on-treatment analysis. The frequencies of complications were compared between groups using the chi-square test or Fisher's exact test. Calculations were performed using STATA v.14 software (StataCorp LLC, College Station, TX, USA).
3.4. Changes of the hand grip value
There were no significant differences in hand grip strength (maximum and mean value) between the larger RA/UA and smaller RA/UA in the used and also in the unused hand at the baseline (pre-procedural measurements) (Table 4).
Based on univariate logistic regression, the change in mean diameter (IxD) by 1 unit (1mm) of the RA or UA was associated with the odds ratio of HSR (average value) at 30-day follow-up. The predictive value of arterial diameter (IxD) for HSR was of borderline statistical significance for combined RA and UA groups. (OR 0.53; 95% CI 0.27 – 1.02; p=0.058). However, separate analyses for RA and UA did not reveal statistical significance.